# An efficient and precise solution-vacuum hybrid batch fabrication of 2D/3D perovskite submodules

Yingping Fan[1,2,5], Zhixiao Qin[3,5], Lei Lu[1], Ni Zhang[1], Yugang Liang[1], Shaowei Wang[1], Wenji Zhan[1], Jiahao Guo[1], Haifei Wang[1], Yuetian Chen [1,4] ✉, Yanfeng Miao [1,4] ✉ & Yixin Zhao [1,2,4] ✉

The quickly processable solution deposition and accurately controllable vacuum deposition are the two competing mainstream fabrication techniques for perovskite films. However, the former may inevitably leave pinholes on film surface and calls for further treatment, the latter exhibits a generally low processing rate. In this work, we develop a solution-vacuum hybrid batch fabrication to precisely deposit nanoscale two-dimensional (2D) capping layer via all-vacuum evaporation on a solution-deposited three-dimensional bulk film. The all-vacuum-deposited 2D perovskite capping layer can be finely controlled with desired composition and stoichiometry to passivate defects and heal the pristine pinholes. We demonstrate the high processing scalability of this solution-vacuum hybrid deposition with the fabrication of 30 cm × 30 cm pinhole-free perovskite submodules, which achieve a champion power conversion efficiency (PCE) up to 22.10% (certified PCE of 21.79%). Our discovery lays out a novel way for efficient and reproducible large-scale production of perovskite modules.

The research on perovskite solar cells (PSCs) has made substantial progress as the latest certified power conversion efficiency (PCE) of small-area, single-junction PSCs has reached up to 27.0%[1]. Therefore, exploiting efficient and reliable technical pathways to promote the practical deployment of perovskite photovoltaics on a large scale has become the inevitable next step, where the engineering prospects of perovskite module fabrication should be scrutinized with the challenges fully addressed.

Post the deposition of perovskite films by solution chemistry methods, the occurrence of pinholes and defects are always unavoidable, which could greatly affect the device efficiency and stability[2–4]. Engineering the perovskite films' top surface is thus essential to ensure the performance and stability of perovskite solar cells[5–8]. In particular, constructing two-dimensional (2D)/three-dimensional (3D) heterojunction in situ could nicely passivate the native defects in a feasible fashion[9–13]. Regularly, the 2D/3D heterojunctions are fabricated by coating the solution of organic salts, such as alkylammonium or phenylammonium halides, onto the as-fabricated 3D perovskite films or 3D perovskite/$PbI_2$ films[14–18]. The surface of 3D perovskite film is then reconstructed into 2D perovskite by ion-exchange reaction, which is simple and effective for defect passivation in small-area samples[19–22]. Unfortunately, when enlarging the device area to the commercial-viable perovskite solar modules (PSMs), there is always a drastic efficiency drop, which can be ascribed to the inextensibility of the regular approach. In detail, during the solution-based post-treatment process, the stoichiometry between

[1]School of Environmental Science and Engineering, Frontiers Science Center for Transformative Molecules, State Key Laboratory of Green Papermaking and Resource Recycling, Shanghai Jiao Tong University, Shanghai 200240, China. [2]Future Photovoltaic Research Center, Global Institute of Future Technology, Shanghai Jiao Tong University (SJTU-GIFT), Shanghai, China. [3]Shanghai Pvsktech Co., Ltd., Shanghai, China. [4]Shanghai Non-carbon Energy Conversion and Utilization Institute, Shanghai, China. [5]These authors contributed equally: Yingping Fan, Zhixiao Qin. ✉e-mail: yuetian.chen@sjtu.edu.cn; yanfengmiao@sjtu.edu.cn; yixin.zhao@sjtu.edu.cn

$PbI_2$ and organic salt in the solution could dictate the formation and conversion of 2D perovskite, which could affect the interface and eventually the device performance[23–25]. Since the ion-exchange reaction is sensitive to the solution concentration and reaction time, the large-area 2D layer formed by this method is usually inhomogeneous on the surface and of varied n values at the interface, which could be integrated into a charge-selective contact stack and impede the carrier transport due to the band misalignment[26–29]. Moreover, such salt-solution method is not adoptable for healing the pinholes on pristine 3D films, especially for those in micrometer-scale. Therefore, achieving a precisely controllable preparation of 2D perovskite layers with suitable chemical composition, uniform n values and thickness for defect passivation and stability enhancement is still a challenging problem for the scaling-up fabrication of perovskite solar modules from both scientific and engineering perspectives.

In this work, we develop an efficient and precise method for solution-vacuum hybrid batch fabrication of 2D/3D perovskite submodules. On a solution-fabricated formamidinium lead iodide ($FAPbI_3$) perovskite film, all-vacuum evaporation of $PbI_2$ and n-hexylammonium bromide (HABr) can accurately deposit a composition-tunable 2D perovskite capping layer to passivate defects and heal pinholes. Based on such a strategy, we demonstrate a high PCE of 25.70% on $FAPbI_3$ PSC of 2D/3D heterojunction with good stability. 30 cm × 30 cm pinhole-free perovskite submodules (aperture area: 663 cm²) are also fabricated by the solution-vacuum hybrid deposition, which achieves a champion PCE up to 22.10% with a certified value of 21.79%, demonstrating nice processing scalability of this method. It is believed that this study could pave the way for precise and effective passivation of large-area perovskite films, which could facilitate the stability and efficiency progression for the large-scale commercial production of perovskite photovoltaics.

## Results

### Precise deposition of 2D perovskite capping layer

As illustrated in Fig. 1, our proposal on the manufacturing process of large-area, high-quality perovskite modules can be divided into three major sections: substrate preparation, 2D/3D perovskite batch

fabrication and electrode fabrication. During the 2D/3D perovskite batch deposition process, fabrication efficiency is assured by the solution deposition method, such as slot-die coating. It usually takes less than a minute to slot-die-coat a perovskite film of 500-600 nm thickness. Such a process could increase productivity and reduce costs compared to the all-vacuum deposition. Yet still, considering the poor accuracy of the solution method for nanometer-thick deposition, vacuum deposition is used to guarantee the precision of the 2D perovskite layer with 10–20 nm thickness. Hence, the proposed perovskite batch fabrication could combine the advantages of fast 3D perovskite fabrication through the solution method with precision control over the 2D passivation layer achieved via vacuum deposition.

Detailed schematic illustrations of the vacuum deposition procedures are shown in Fig. 2a with scanning electron microscopy (SEM) images for each of the corresponding stages attached below in Fig. 2. As correlated between Step 1 in Fig. 2a and the SEM image in Fig. 2b, the solution-deposited pristine $FAPbI_3$ perovskite films exhibit pinholes and defects[30,31], where the pinholes are of nano- and micron-size distribution. During the following vacuum evaporation process, $PbI_2$ layer was deposited onto the $FAPbI_3$ perovskite film with controlled thickness as the photo shown in Supplementary Fig. 1 and characterized in Fig. 2[32]. The morphological results in Fig. 2c reveal that pinholes can still be observed on the surface of $FAPbI_3$-$PbI_2$ film, and there is no significant change in the grain size between $FAPbI_3$ and $FAPbI_3$-$PbI_2$ films[33]. The measurement based on the high-resolution transmission electron microscopy (HR-TEM) image in Fig. 2c inset indicates that the thickness of the $PbI_2$ layer is ~7 nm. The 2D perovskite capping layer was then fabricated by subsequently vacuum evaporating HABr and thermal annealing (Fig. 2a). The reaction between HABr and $PbI_2$ with different molar ratios (HABr: $PbI_2$ = x: 1, x = 2, 3, and 4) will lead to the formation of different 2D perovskites, such as $HA_2PbI_2Br_2$ and $HA_2FAPb_2I_5Br_2$ with good intrinsic stability and nice surface coverage. SEM images of this 2D capped sample in Fig. 2d show that some layered structures formed on top of $FAPbI_3$ perovskite films, with the previous pinholes disappeared (Supplementary Fig. 2), suggesting the pinhole-healing effect of this deposition process. The cross-sectional SEM images and atomic force microscopy (AFM) images reveal that

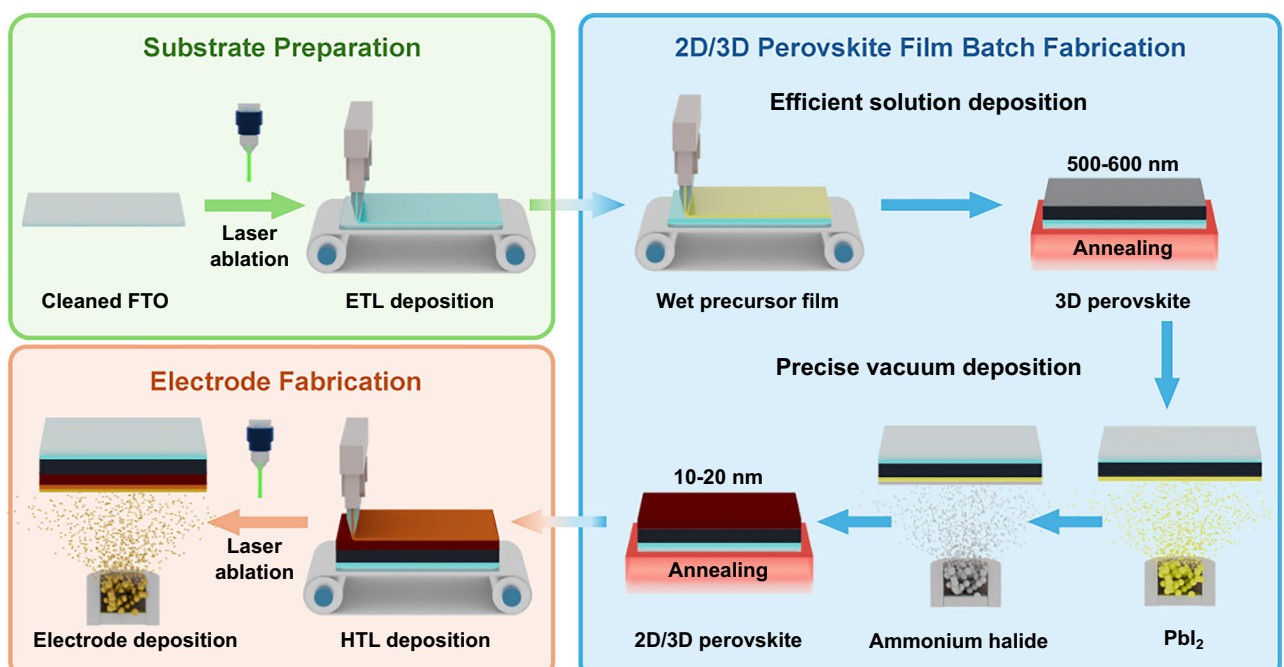

**Fig. 1 | Manufacturing process of perovskite modules.** Schematics of the proposed solution-vacuum hybrid perovskite submodule manufacturing process. FTO fluorine-doped tin oxide, ETL electron transport layer, and HTL hole transport layer.

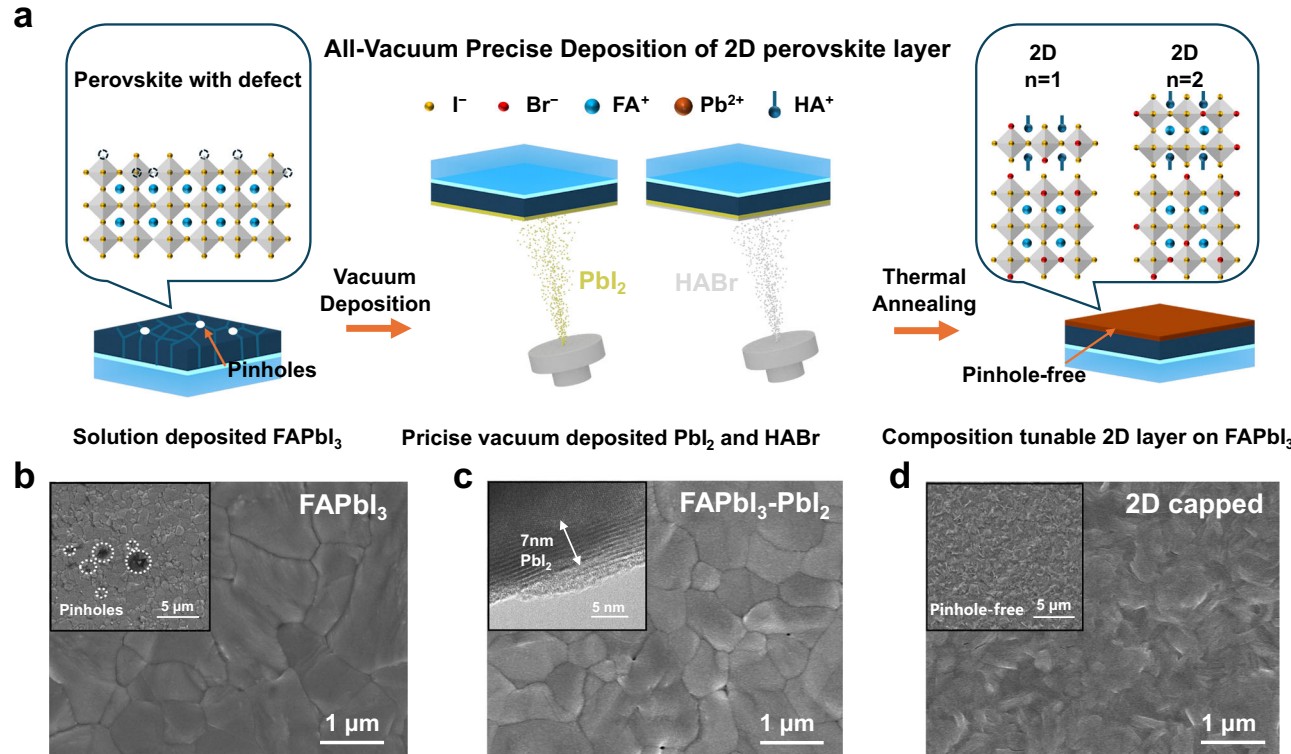

**Fig. 2 | Morphology changes during the vacuum evaporation process. a** Detailed schematic diagram of precise vacuum deposition for the 2D perovskite layer process. **b**–**d** Scanning electron microscopy (SEM) images of **b** FAPbI$_3$, **c** FAPbI$_3$-PbI$_2$, and **d** 2D capped samples. Insets are the corresponding larger-area SEM or high-resolution transmission electron microscopy (HR-TEM) images.

both the FAPbI$_3$ and 2D capped samples exhibit similar roughness (Supplementary Figs. 2, 3), indicating that the 2D capping layer has a good coverage on 3D FAPbI$_3$ films.

Considering that the stoichiometric ratio between HABr and PbI$_2$ may have an impact on the type and properties of the formed 2D perovskites, 2D/3D films were subsequently prepared by evaporating different molar ratios of HABr and PbI$_2$. These samples are herein denoted as HABr: PbI$_2$ = x: 1, in which x is 0, 2, 3, 4, respectively. Grazing-incidence wide-angle X-ray scattering (GIWAXS) patterns are collected to analyze the composition of different 2D/3D films as shown in Fig. 3 and Supplementary Fig. 4. In x = 0 sample, the diffraction ring of PbI$_2$ signal is observed, which is consistent with the XRD patterns in Supplementary Fig. 5. While a sharp diffraction ring emerged at scattering vector q = 3.9 nm$^{-1}$ in the GIWAXS pattern of x = 2 sample in Fig. 3a. This diffraction ring can be indexed as the (002) plane of n = 1 perovskite, demonstrating that HABr directly reacted with evaporated PbI$_2$ and led to the formation of HA$_2$PbI$_2$Br$_2$ perovskite (Supplementary Figs. 5, 6)[34,35]. The GIWAXS pattern of x = 3 film in Fig. 3b exhibits two characteristic diffraction rings, indicating that two types of 2D perovskites as n = 1 and n = 2 (q = 3.0 nm$^{-1}$) were formed. When increasing the ratio to x = 4, there is only one diffraction ring of the n = 2 2D perovskite as shown in Fig. 3c, which is also consistent with the XRD results in Supplementary Figs. 6, 7. Hence, the exact chemical composition and n value of the 2D perovskites can be precisely controlled by adjusting the ratio of HABr to PbI$_2$, which was further corroborated by the crystal structures identified in the HR-TEM images (Fig. 3d) and PL spectra (Supplementary Fig. 8) for these samples[36]. The SEM images in Supplementary Fig. 9 also reveal that the surface morphology of the FAPbI$_3$ films post-treatment is different as the molar ratio of HABr to PbI$_2$ increases from 2 to 4.

To investigate the transformation of 2D/3D heterojunctions during annealing, the effect of annealing time on the crystal structure was studied as the XRD patterns shown in Supplementary Fig. 10. In

the case of x = 2, the peak position of the characteristic 2D perovskite has almost no change. While for x = 3 (Supplementary Fig. 10b) and x = 4 (Supplementary Fig. 10c) samples, the characteristic peaks' positions show significant changes along annealing. The real-time changes for the x = 3 and x = 4 samples were further monitored by in situ GIWAXS measurement. For x = 3 sample (Fig. 3e), a diffraction peak of n = 1 2D perovskite emerged at the beginning of the annealing process. As the annealing proceeded, the signal intensity of n = 1 2D perovskite gradually weakened, accompanied by the gradual increment in the peak strength of n = 2 perovskite. This result indicated that the 2D perovskites in x = 3 sample undergo a transformation from n = 1 to n = 2 2D perovskite. For x = 4 sample, as shown in Supplementary Fig. 11, there is another new diffraction peak emerged at the scattering vector q = 3.4 nm$^{-1}$ that corresponds to HABr (Supplementary Fig. 12), indicating that there is an excessive layer of HABr covered on the surface. It can be found that during the annealing process, the excessive HABr and n = 1 2D perovskite can both be converted to n = 2 2D perovskite.

The possible formation and conversion mechanism of the 2D perovskite capping layer with different molar ratios of HABr and PbI$_2$ are schemed out in Supplementary Fig. 13. When x = 2, the deposited HABr would undergo solid phase reaction with PbI$_2$ to form n = 1 2D perovskite and exhibit no change over the subsequent annealing process. When x = 3, a 2D perovskite with n = 1 is initially formed. But after annealing, the n = 1 2D perovskite at the interface between 3D perovskite and 2D perovskite capping layer is converted into n = 2 2D perovskite by the progression of solid-state reaction. Further, when x = 4, residual HABr and n = 1 2D perovskites are all converted into n = 2 perovskites under the thermal stress during annealing. Therefore, it can be concluded that for the proposed technical route, the conversion process of n = 1 and n = 2 2D perovskites can be precisely controlled by adjusting the molar ratio between HABr and PbI$_2$.

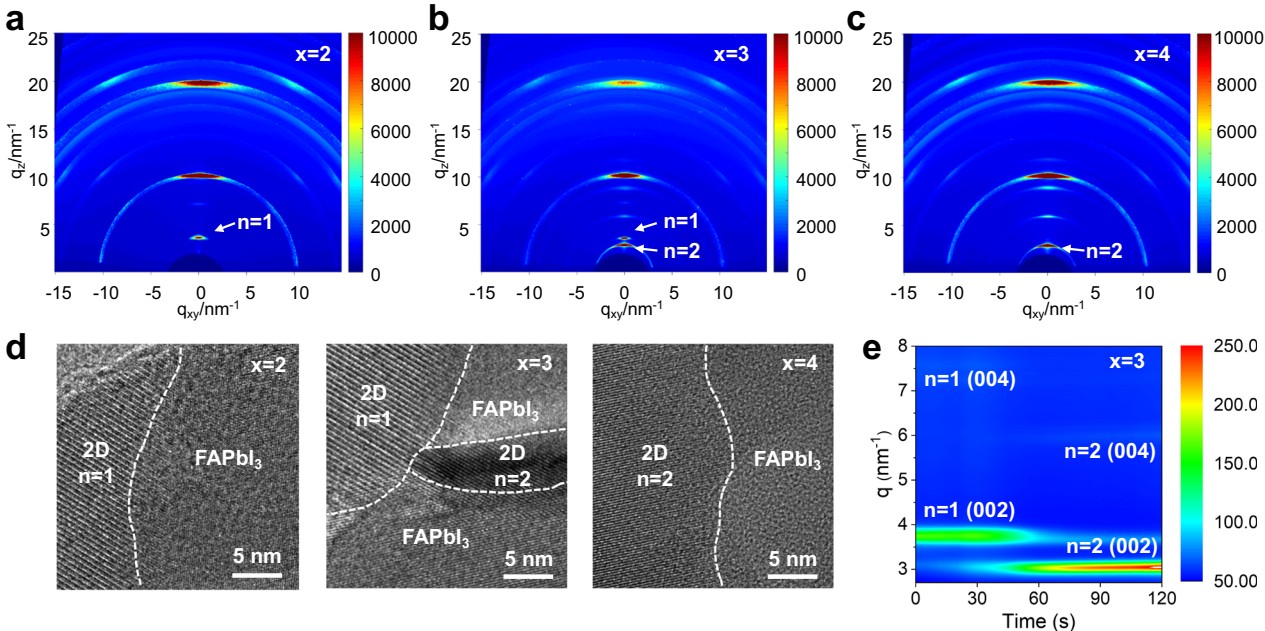

**Fig. 3 | Formation and conversion process of 2D perovskite. a–c** Grazing-incidence wide-angle X-ray scattering (GIWAXS) patterns of post-treated perovskite films with HABr: $PbI_2 = x : 1$, where **a** $x = 2$, **b** $x = 3$, and **c** $x = 4$. **d** HR-TEM images of the $x = 2, 3,$ and 4 film samples. **e** In situ GIWAXS patterns of $x = 3$ sample.

## Characterization of 2D perovskite-capped films

Photoluminescence (PL) mapping was profiled to evaluate the quality of the evaporated 2D perovskite layer on $FAPbI_3$ perovskite films. Supplementary Fig. 14 shows that the control $FAPbI_3$ and $x = 0$ films have relatively low PL intensities, suggesting that there are many defects on the film surface. In the case of $x = 2-4$ samples (Fig. 4a), the residual $PbI_2$ has been completely converted into 2D perovskites, and the PL intensity of the films becomes much stronger, showing an effective passivation effect. In particular, the PL intensity of $x = 3$ sample is more uniform than $x = 2$ and $x = 4$ samples. In the $x = 4$ sample, irregular areas with lower PL intensity were observed, which corresponds to the rougher 2D perovskite capping layer (Supplementary Fig. 9c). Time-resolved photoluminescence (TRPL) spectra in Supplementary Fig. 15 and Supplementary Table 1 show that, the PL lifetimes of $x = 3$ and $x = 4$ samples are much longer than that of other samples. The enhancement of PL intensity and the prolongation of carrier lifetime is an indication of the defect passivation enabled by the 2D perovskite capping[37,38].

Kelvin probe force microscopy (KPFM) was carried out to characterize changes in the surface potential of different samples. As shown in Supplementary Fig. 16, the surface potential of $x = 0$ decreases compared to the $FAPbI_3$ film. As $PbI_2$ converts to 2D perovskite, the general surface potential in $x = 2-4$ samples (Fig. 4b) decreases at least 250 mV compared to the $FAPbI_3$ film, indicating that the formation of 2D perovskite at the interface could greatly facilitate the transport of holes from 3D perovskite to the hole transport layer (HTL)[39]. Especially, the profiles of the film surface potential of $x = 3$ sample tend to be more homogenized and lower than $x = 2$ and $x = 4$ samples, indicating that the recombination on surface and at the grain boundaries of the perovskite film have been well suppressed[40].

To further investigate the effect of the 2D perovskite capping layer on carrier transport, cross-sectional KPFM mapping was performed to detect the charge-carrier extraction barriers at the perovskite/HTL interface. The Δ electrical field was calculated by the first derivative of potential. Supplementary Fig. 17 shows that the electrical field at the perovskite/HTL interface is slightly enhanced for the $x = 0$ sample, suggesting the reduction of the interfacial defect density[41,42]. The electrical field at the interface further increases in $x = 2-4$ samples (Fig. 4c), and there is a tendency to move into the bulk phase of the perovskite, which could more effectively repel electrons[43]. Stronger electric field differences after 2D perovskite insertion between the perovskite/HTL contact further prove a lower interfacial defect density.

Besides surface passivation, the precise deposition of 2D perovskites has great applicational potential for pinhole repairment, which addresses a practical issue that cannot be achieved by the commonly used alkylammonium halide post-treatment. An optical microscope was used to observe the changes in pinholes before and after 2D perovskite deposition. Figure 4d shows that, in the $FAPbI_3$ sample, there is a pinhole in sight with several micrometers in diameter. After 2D perovskite evaporation, the pinhole was covered with flake-like $n = 1$ 2D perovskite. Moreover, comparison for the PL mapping image of the $x = 3$ sample in Fig. 4e shows significantly reduced PL quenching for areas around the pinhole, presenting a great advantage for this hybrid method to comprehensively improve the film quality and realize efficient photovoltaic devices.

## Device performance and characterization

With the device configuration of $FTO/TiO_2/SnO_2/perovskite/Spiro-OMeTAD/Au$, the photovoltaic performance of small-area solar cells after the formation of the 2D perovskite capping layer and different HABr to $PbI_2$ molar ratios were compared in Fig. 5a and Supplementary Table 2. Open-circuit voltage ($V_{oc}$), current density ($J_{sc}$), and fill factor (FF) values of $x = 2$ and $x = 3$ samples are greatly enhanced compared to that of the samples without HABr deposition. However, for the $x = 4$ sample, $V_{oc}, J_{sc}$, and FF of the device have drastically decreased. The thermally deposited $n = 2$ 2D perovskite layer, while demonstrating effective passivation of surface defects in perovskite materials, exhibits a detrimental impact on device performance due to excessive overlayer thickness, which significantly impedes hole transport and consequently leads to a substantial reduction in device efficiency[20]. Which may be due to the carrier transport problem caused by an excessively thick 2D capping layer. Hence, in the following research, $x = 3$ devices are selected as the target group (denoted as vacuum) and pristine $FAPbI_3$ devices are the control group. Supplementary Fig. 18 shows the forward/reverse scan $J$-$V$ curves and statistical PCEs of

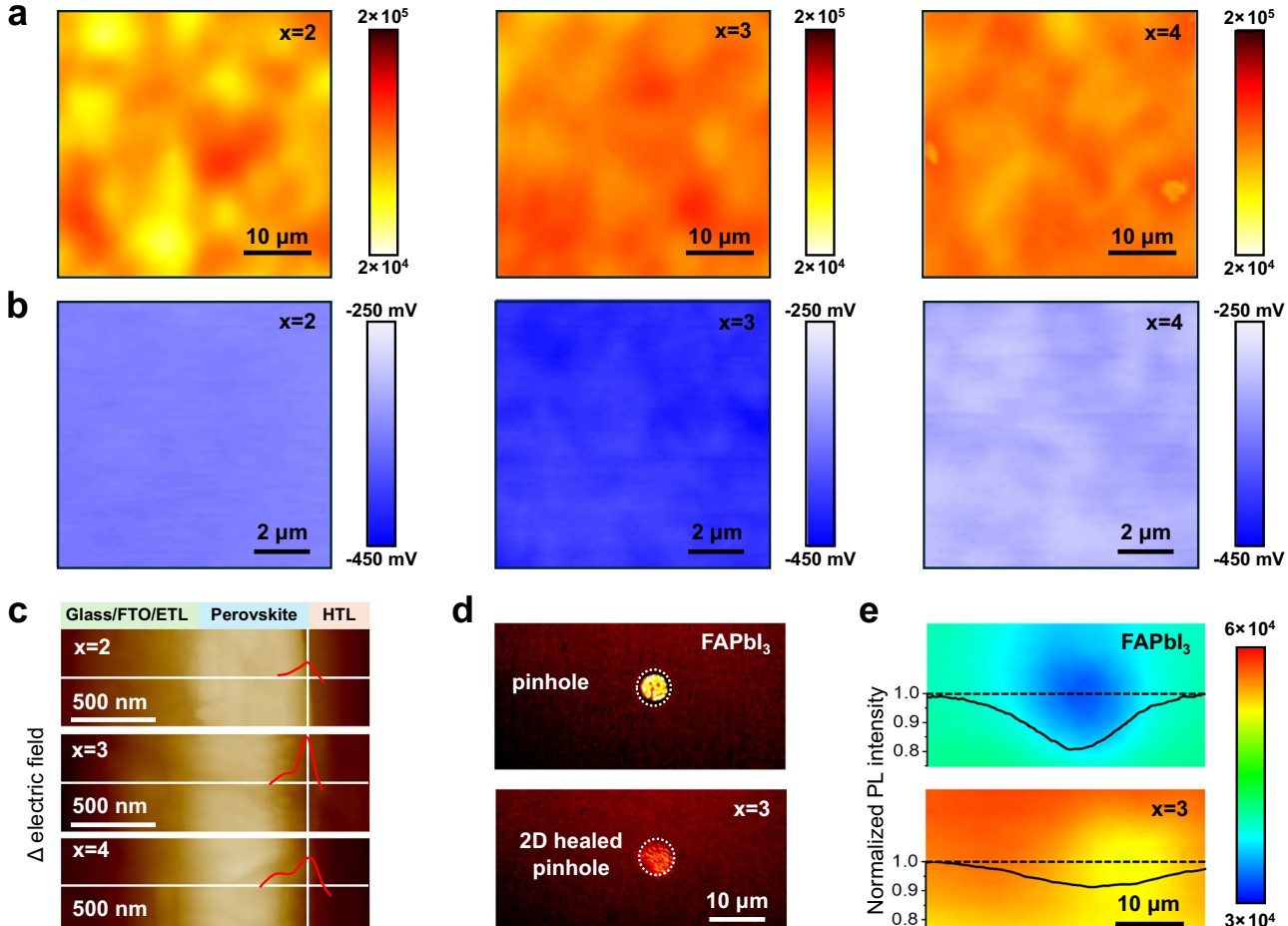

**Fig. 4 | Characterization of 2D perovskite-capped films. a** Photoluminescence (PL) mapping profiles and **b** Kelvin probe force microscopy (KPFM) images of perovskite films with different molar ratios of HABr to $PbI_2$. **c** Electric field distribution at the interface between perovskite and hole transport layer (HTL) acquired via KPFM cross-sectional scans. **d** Optical microscope images of $FAPbI_3$ and x = 3 samples. **e** Comparison on PL mapping of the $FAPbI_3$ and x = 3 films. The curves represent the normalized PL intensity for the $FAPbI_3$ and x = 3 samples.

control and vacuum PSCs. In detail, the control device exhibited a power conversion efficiency (PCE) of 22.45% with a $V_{oc}$ of 1.110 V, $J_{sc}$ of 25.17 mA cm$^{-2}$ and FF of 80.34%. The vacuum cell achieved a champion PCE of 25.70% with a $V_{oc}$ of 1.165 V, $J_{sc}$ of 25.98 mA cm$^{-2}$ and FF of 84.92%, which delivered generally enhanced $V_{oc}$, FF and reduced hysteresis. External quantum efficiency (EQE) spectra and integrated current density in Supplementary Fig. 19 also matched well with the measured $J_{sc}$. Steady-state output results demonstrate that the 2D capping layer exhibits excellent protective performance under continuous illumination operating conditions of the device, as shown in Supplementary Fig. 20.

The underlying physical and electrical reasons for the enhancement of photovoltaic parameters have been thoroughly studied with a combination of characterization techniques. According to the reciprocity theorem, a low $V_{oc}$ loss should lead to a high electroluminescence (EL) efficiency. As confirmed in Fig. 5b, the vacuum device exhibited an external quantum efficiency of electroluminescence ($EQE_{EL}$) of up to 8.64% at an injected current density of $J_{sc}$, suggesting reduced interfacial non-radiative recombination[44]. Mott–Schottky plots in Supplementary Fig. 21 show that the vacuum device reveals a slightly higher (1.00 V) built-in potential ($V_{bi}$) than the control device (0.94 V). Results based on electrochemical impedance spectroscopy (EIS, Supplementary Fig. 22) indicate that the vacuum device has higher recombination resistance. The space-charge limited current (SCLC) results in Supplementary Fig. 23 confirms that the hole-only device based on the vacuum sample has a lower trap density[45].

The dark current measurements also indicate lower leakage current of the vacuum device (Supplementary Fig. 24). Figure 5c illustrates that the vacuum device displays a lower ideal factor (n) value and series resistance ($R_s$). Considering that the $EQE_{EL}$ have confirmed that the interface recombination is largely suppressed, the lower n value demonstrates a lower recombination process in the vacuum devices[44,46]. Meanwhile, in Supplementary Fig. 25, saturated recombination current density ($J_O$) of the vacuum device (2.23 × 10$^{-12}$ mA cm$^{-2}$) is lower than that of the control one (2.23 × 10$^{-9}$ mA cm$^{-2}$), indicating suppressed non-radiative recombination for higher $V_{oc}$. All the above results prove that the 2D capping layer could improve the charge-carrier transport capacity, reduce trap states and suppress the non-radiative recombination. Quasi-PSCs were fabricated for conductive atomic force microscopy (c-AFM) measurement with the device structure of FTO/TiO$_2$/SnO$_2$/perovskite/Spiro-OMeTAD, where the c-AFM tip acted as the top electrode of the quasi-PSC. As shown in Fig. 5d, the vacuum quasi-PSC exhibits increased current and a more homogeneous distribution, indicating efficient charge extraction, which is attractive for achieving higher FF and $J_{sc}$.

Depth-dependent ultraviolet photoelectron spectroscopy (UPS) measurement was further performed to understand how the 2D perovskite at the interface modifies the electronic band structure. Figure 5e shows the profiles of valence band maximum (VBM) as a function of depth for the vacuum sample, which was extracted from the original spectra in Supplementary Fig. 26, and the corresponding parameters are listed in Supplementary Table 3. The magnitude of band bending at

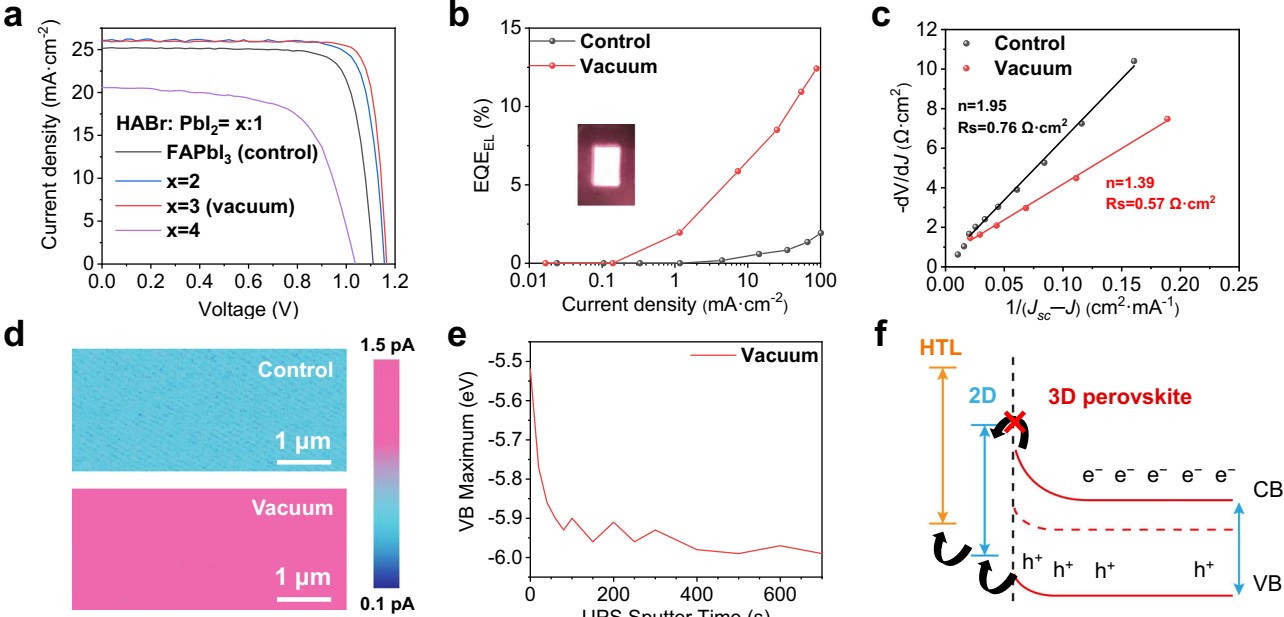

**Fig. 5 | Device performance and enhanced band alignment. a** *J-V* curves of the best FAPbI₃(control), x = 2, x = 3(vacuum) and x = 4 devices under reverse scans. **b** External quantum efficiency of electroluminescence (EQE$_{EL}$) measurements of the control and vacuum perovskite solar cells (PSCs). **c** Plots of $-dV/dJ$ with respect to $1/(J_{sc} - J)$ of control and vacuum devices. **d** Conductive atomic force microscopy (c-AFM) images of control and vacuum films. **e** Valence band maximum determined from depth-dependent ultraviolet photoelectron spectroscopy (UPS) of the vacuum sample. **f** Schematic illustration of the energy band alignment of the vacuum sample. Source data are provided as Source Data file 1.

the surface increases, which is consistent with previously reported UPS studies[47]. Fig. 5f schemes out the energy level alignment diagram between the perovskite and the HTL. Specifically, the vacuum deposition process leads to a wider bandgap at a previously defective surface and a larger overall band bending that extends deep into the bulk sample. The compositional gradients and surface energetics can be precisely tuned. In contrast, the VBM in the control sample did not show significant changes, as shown in Supplementary Fig. 27 and Table 4. Moreover, the ideal energy level alignment allows for efficient charge transfer at the hole-selective heterointerface, while the shallower conduction band minimum (CBM) of the vacuum sample can block electron transport and thus reduce electron-hole recombination at the interface. As illustrated in Supplementary Fig. 28, depth-dependent X-ray photoemission spectroscopy (XPS) shows that the atomic percentage of C in the thin film from surface to the bulk exhibits a notable decrease from 50 to 20%, indicating a gradient distribution pattern of 2D perovskite. Within the bulk phase of the perovskite, the atomic ratio of C to I is approximately 1:3, suggesting that the bulk primarily consists of FAPbI₃ perovskite.

The reduction in defect density could also facilitate good retainment on the device's operational stability. Moisture stability tests were performed to evaluate the protection effect of the 2D perovskite capping layer. After being stored under 80% relative humidity (RH), 25 °C in ambient conditions for one day, the control sample had undergone the phase change from α- to δ-FAPbI₃ as shown in Supplementary Fig. 29. In contrast, the area covered with 2D perovskite capping layer remained black phase, which revealed that the 2D layer exhibited excellent protection to the FAPbI₃ perovskite.

For device stability, the shelf life was measured by storing the unencapsulated devices in dark at 25 °C and in an ambient air glovebox with 25% RH. Supplementary Fig. 30 demonstrates a 20% decrease in the PCE of the control device after 1000 h of aging (ISOS-D-1), while the vacuum device retains 95.5% of its initial PCE. The operational stability of the PSCs was further investigated by aging the encapsulated devices in ambient air and using MPP tracking under a temperature of 65 °C

(ISOS-L-2). Figure 6a shows the PCE evolution of the PSCs under such a process. The PCE of the control device dropped to 60% of the initial value after 400 h. By comparison, the PCE of the vacuum cell remained at ~90% of its initial PCE after 1000 h MPP tracking. The MPP results also proved that the construction of a 2D perovskite capping layer by vacuum thermal evaporation has a good passivation and protection effect.

To validate the potential of our solution and vacuum hybrid deposition method for large-area modules, 30 cm × 30 cm PSMs were fabricated. As displayed in Fig. 6b, the best PSM prepared by the solution and vacuum hybrid deposition method achieved a PCE of 22.10% with an $I_{sc}$ of 303.01 mA, a $V_{oc}$ of 59.01 V, and an FF of 81.96%. The vacuum PSM also obtained a certified PCE of 21.79% as the report attached in Supplementary Fig. 31, demonstrating high process scalability of the solution-vacuum hybrid batch fabrication method.

## Discussion

Overall, we have developed an efficient and precise solution-vacuum hybrid batch fabrication strategy to fabricate pinhole-free and high-quality 2D/3D perovskite submodules with high efficiency and good stability. This strategy combines the rapid preparation of 3D perovskite film by solution method (slot-die coating for module fabrication) and precise deposition of 2D perovskite passivation layer by vacuum evaporation, i.e., a highly repeatable and controllable nanoscale 2D perovskite layer can be vacuum-deposited on a solution-prepared 3D perovskite film. As the vacuum-deposited HABr can nicely convert the evaporated PbI₂ into stable 2D perovskite, the chemical composition and proportion of the nanoscale 2D perovskites capping layer can be precisely controlled in the vacuum process by altering the precursors' stoichiometry, where an optimal recipe can be identified for reduction on defected sites and facilitation on interfacial carrier transport. Therefore, the vacuum processing could homogeneously heal pinholes and passivate trap states on solution-fabricated large-area perovskite films, resulting in a certified PCE of 21.79% of a 30 cm × 30 cm pinhole-free perovskite submodule. This study could

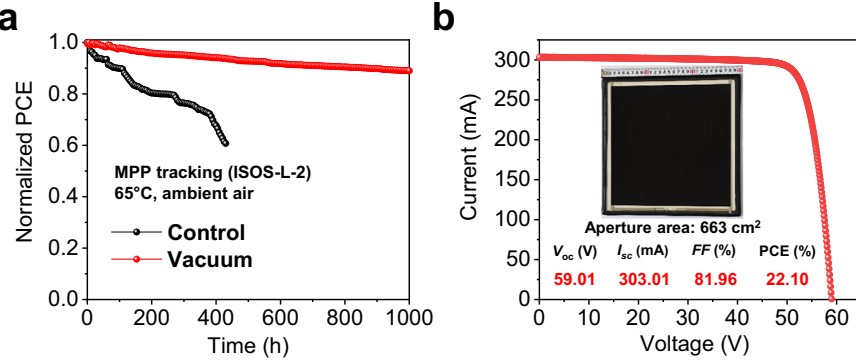

**Fig. 6 | Stability and perovskite module performance. a** Long-term operational stability of control and vacuum devices. **b** *J-V* curve for the vacuum perovskite solar module (PSM) with an aperture area of 663 cm². Inset, the photo image of a PSM. Source data are provided as Source Data file 2.

provide a feasible and programmable technical route for the stream-lined fabrication and practical promotion of high-efficiency, large-area perovskite photovoltaics.

## Methods

### Materials

Lead iodide ($PbI_2$, 99.99%) was purchased from TCI. Tin (IV) oxide colloid precursor ($SnO_2$, 12% in $H_2O$ colloidal dispersion), formamidinium iodide (FAI) and methylamine hydrochloride (MACl, 99%) were purchased from Xi'an Polymer Light Technology Corp. n-Hexylammonium bromide (HABr) was purchased from Great Cell Solar Materials. Tetrakis (dimethylamino) titanium (TDMAT), *N*, *N*-dimethylformamide (DMF, anhydrous, 99.8%), and dimethyl sulfoxide (DMSO, anhydrous, 99.7%) were purchased from J&K Scientific Ltd. Diethyl ether (anhydrous) and chloroform were purchased from Sinopharm Chemical Reagent Co., Ltd.

### Fabrication of small-area PSCs

A compact $TiO_2$ layer of 10 nm was deposited by the atomic layer deposition (ALD) method. The deposition materials were TDMAT and water, the deposition temperature was 150 °C, and the deposition cycle number was 250 cycles. The deposited substrate was annealed at 500 °C for 30 min. The $SnO_2$ solution with colloid precursor/deionized water of a 1:5.5 volume ratio was spin-coated on the $TiO_2$ layer at 3000 rpm for 30 s, followed by annealing at 180 °C for 30 min. The 1.5 M $FAPbI_3$ perovskite precursor was prepared by mixing 1.5 mmol $PbI_2$, 1.5 mmol FAI, and 0.525 mmol MACl in 1 mL mix solvent (889 μL DMF and 111 μL DMSO). The perovskite film was fabricated in a dry-air box with ~25% relative humidity. The perovskite films were deposited on $FTO/TiO_2/SnO_2$ substrate by spin-coating at 5000 rpm for 30 s. During spin-coating, 600 μL diethyl ether was dripped for 10 s before the end of spin-coating. Perovskite films were annealed at 150 °C for 10 min. For 2D perovskite capping layers, $PbI_2$ was vacuum evaporated at a rate of 0.5 Å s⁻¹ to get a thickness of 7 nm under a pressure condition of less than 10⁻³ Pa. HABr (6, 9, and 12 nm) was subsequently vacuum evaporated at a rate of 0.3 Å s⁻¹ under a pressure condition of less than 10⁻³ Pa. After the evaporation process, the films were annealed at 100 °C for 2 min. The Spiro-OMeTAD solution was prepared by mixing 90 mg Spiro-OMeTAD in 1 mL chlorobenzene with 39.5 μL 4-tert-butylpyridine (TBP) and 23 μL Li-bis (trifluoromethanesulfonyl) imide (Li-TFSI)/acetonitrile (520 mg mL⁻¹), and further deposited by spin-coating the solution at 3000 rpm for 30 s. Finally, 80-nm-thick Au was thermally evaporated on the Spiro-OMeTAD layer.

### Fabrication of large-area submodules

Firstly, a picosecond laser scriber was used to ablate the pattern of P1 on FTO. Then the $TiO_2$ and $SnO_2$ layers were deposited by ALD and slot-die coating, respectively. Slot-die coating (nTact, nRad2) of the

$SnO_2$ colloid precursor solution was done on the $FTO/TiO_2$ substrate (coating speed: 10 mm s⁻¹, solution supply rate: 10 μL s⁻¹, gap between the substrate and the slot-die lip: 80 μm). Then the film was annealing at 150 °C for 30 min. For perovskite deposition, perovskite precursor solution (1.2 M) was slot-die coated, followed by a vacuum-flashing process. Then the film was annealing at 150 °C for 10 min. The coating speed was 20 mm s⁻¹, the solution supply rate was 20 μL s⁻¹, and the gap between the substrate and the slot-die lip was 110 μm. For the 2D perovskite capping layer, at the first step, $PbI_2$ were vacuum thermally evaporated at an evaporation rate of 0.5 Å s⁻¹. HABr is subsequently vacuum thermal evaporated with the rate of 0.3 Å s⁻¹. After the evaporation process, the film was annealed at 100 °C for 2 min. Spiro-OMeTAD was also slot-die coated on the perovskite films. The coating speed, solution supply rate and the gap between the substrate and the slot-die lip were 20 mm s⁻¹, 15 μL s⁻¹, and 60 μm, respectively. The P2 lines were then applied near the P1 lines by laser scribing, and finally, the P3 lines were scribed after the deposition of the Au electrode. The widths of P1, P2, and P3 are 13, 54, and 18 μm, respectively. The distances between P1-P2 and P2-P3 are 40 and 30 μm, respectively. The geometrical fill factor is ~97%.

### Characterization

The XRD patterns of perovskite films were characterized by a Shimadzu XRD-6100 diffractometer with Cu Kα radiation. UV-vis spectra were collected on a Cary-60 UV-Vis spectrophotometer. TRPL spectra were measured at room temperature on an FLS1000 photoluminescence spectrometer (Edinburgh Instruments Ltd.) under a 450 nm excitation laser. PL mapping was measured by Renishaw inVia confocal Raman microscope. The film morphology was characterized by a SEM (JEOL JSM-7800F Prime). HR-TEM was measured by Talos F200X G2, and samples were prepared by scraping the perovskite films from the glass substrate and dispersing them in chlorobenzene. GIWAXS measurements were performed at the BL14B1 beamline of the Shanghai Synchrotron Radiation Facility (SSRF) with a beam wavelength of 0.12398 nm. Kelvin probe force microscopy (KPFM) was performed on perovskite samples using a Dimension FastScan Bio Atomic Force Microscope (Bruker) under illumination. For UPS depth profiling, the perovskite thin film samples were loaded into a Kratos Axis Supra under ultrahigh vacuum (10⁻⁷–10⁻⁸ Torr). *J-V* curves of perovskite solar cells were measured under an inert atmosphere by a Keithley 2401 source meter with a scan rate of 20 mV s⁻¹ under simulated AM 1.5G illumination (100 mW cm⁻²; Enlitech Class AAA Solar Simulator). EQE was measured on an Enlitech QE-3011 system. The $EQE_{EL}$ was measured under an inert atmosphere by a Keithley 2400 source meter and a fiber spectrometer (QE 65 Pro, Ocean Optics). The dark *I-V* for SCLC measurements were performed on devices with the structure of $FTO/c-TiO_2/perovskite/PCBM/Ag$ using a Keithley 2401 source meter in the dark.

## Reporting summary

Further information on research design is available in the Nature Portfolio Reporting Summary linked to this article.

## Data availability

The data that support the findings of this study are available in the paper and Supplementary Information. Source data are provided with this paper.

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

## Acknowledgements

This work is supported by the National Natural Science Foundation of China (NSFC, Nos. 22025505, 22220102002, 52203334, 22209111, and 22479098), the Natural Science Foundation of Shanghai (Nos. 23ZR1432300 and 23ZR1428000), and the Oceanic Interdisciplinary Program of Shanghai Jiao Tong University (grant no. SL2022ZD105). The authors gratefully thank the Shanghai Synchrotron Radiation Facility of CAS for assistance on GIWAXS and in situ GIWAXS measurements. The authors also thank the School of Environmental Science and Engineering and the Instrumental Analysis Centers of Shanghai Jiao Tong University for assistance with material characterizations.

## Author contributions

Y.Z. conceptualized the research. Y.Z., Y.M., and Y.C. supervised the research. Y.F. prepared PSCs and film samples for characterizations and data analysis. Z.Q., L.L., and N.Z. carried out the fabrication and characterization of PSMs. Y.L. and S.W. assisted with the PL mapping, GIWAXS and HR-TEM measurements. W.Z. assisted with TRPL and $EQE_{EL}$ characterizations. J.G. and H.W. participated in AFM and MPP characterizations. Y.Z., Y.M., Y.C., and Y.F. wrote and edited the paper with input from all authors.

## Competing interests

The authors declare no competing interests.
