## [Transparent Peer Review file · Nature Communications]

An Efficient and Precise Solution-Vacuum Hybrid Batch Fabrication of 2D/3D Perovskite Submodules

Corresponding Author: Professor Yixin Zhao

Version 0:

Reviewer comments:

Reviewer #1

(Remarks to the Author)

I co-reviewed this manuscript with one of the reviewers who provided the listed reports as part of the Nature Communications initiative to facilitate training in peer review and appropriate recognition for co-reviewers.

In this study, the authors utilized a sequential vacuum deposition method to fabricate a 2D perovskite capping layer through a two-step evaporation process involving HABr and Pbl₂. The adjustment of the molar ratios (HABr/Pbl₂ = x/1) influenced the dimensionality of the 2D layer, and the subsequent investigation was pivotal in determining the optimal ratio. This optimization led to a lower interfacial defect density and a reduction in recombination, leading to the best device performance observed at x = 3. Furthermore, the authors successfully upscaled the stack to a PSM with an active area of 30x30 cm². The work is interesting, and the achieved efficiency on the PSM is remarkable. However, similar ideas, yet with hybrid approaches, was already presented in literature (Wen, J., Zhao, Y., Wu, P. et al. Heterojunction formed via 3D-to-2D perovskite conversion for photostable wide-bandgap perovskite solar cells. Nat Commun 14, 7118 (2023). <https://doi.org/10.1038/s41467-023-43016-5>). In order to further consider the manuscript, the analysis performed need improvements and the work requires major revisions.

Below are some specific comments on the manuscript:

General Comments:

- Authors need to discuss better the novelty of the paper compared to the on of Wen J. et al discussed above.
- Stability test is only limited to light soaking, however a full test at 85 °C is required considering that the use of SPIRO-OMeTAD is very limiting for thermal stress.
- Ensure the manuscript contains adequate references and proper contextualization within the existing literature.

Specific comments:

- Line 38: The new record in the NREL chart is 27.0% (on small area, about 0.05 cm²). Please update this value.
- Line 86: Do the authors mean "deposit" with "scrape-coat"? Please modify.
- Line 106: Please specify the chemical formula of the 2D perovskite layer for clarity.
- Line 116: The authors mention the evaporation of various molar ratios of HABr to Pbl₂. Did they verify these molar ratios using EDX analysis, or were they assumed based on evaporation rates/thicknesses? Please clarify.
- Figure 3d: The formation of the 2D perovskite layer appears non-uniform over the 3D structure. Literature has shown that 2D perovskites can either form a conformal coating over the underlying layer or exhibit discontinuities. Please address this point explicitly and clarify the reason behind this significant lack of uniformity.
- Ensure that the sources or databases used for obtaining the XRD peak references are clearly specified.
- In Supplementary Fig. 7, the XRD spectra for samples with 2D capping layers across various annealing durations are presented. The final two annealing times for x=2 and x=4 exhibit similar spectra, indicating a stable phase of the 2D has been achieved. However, for x=3, some changes occur in the plot between 60 and 90 seconds. Furthermore, the authors state in the experimental section that devices were annealed for 2 minutes. Therefore, the XRD pattern does not directly correlate with the conditions used for device fabrication. It is advised to include the XRD pattern up to 2 minutes for x=3.
- Line 215-216: The poor performance of the device with x=4 is attributed to excessive thickness of the 2D capping layer in this case. However, the previous analysis has shown a lower electric field, with a less intense peak compared with x=3, at the PVK/HTL interface. This analysis also showed reduced uniformity and higher surface potential. The authors should consider and correlate with the previous analysis when discussing the electrical characteristics.

- 9) Line 238: A lower ideality factor is not always indicative of improved device performance, as it depends on the nature of the recombination processes occurring in the device (doi: 10.1002/aenm.202000502). Please consider that.
- 10) Line 263: Correct the figure reference from "Fig. 4f" to "Fig. 5f".
- 11) Lines 264-265: The discussion on energy level alignment is fragmented, continuing onto line 269. It would be more coherent to consolidate this section into a unified discussion. Additionally, it is recommended to include UPS measurements for the control device for a more comprehensive analysis.
- 12) Line 266: The conclusions drawn from the XPS analysis are unclear. Please clarify the key findings and their significance.
- 13) The inset in Supplementary Fig.24 is not clearly visible and indistinguishable. Please improve its visibility and clarity.
- 14) Line 299: Specify large area since slot-die coating was employed only for module fabrication.
- 15) Line 31: 30x30 cm² is a small module size, not a submodule, as reported in doi: 10.1002/pip.3489. Please modify that.

Reviewer #2

(Remarks to the Author)

In this manuscript, Fan et al. report an efficient and precise solution-vacuum deposition strategy that enables the rapid and controlled fabrication of 2D/3D perovskite submodules. The vacuum-deposited 2D capping layers can be finely tuned by adjusting the molar ratio of PbI₂ and HABr. As a result, the authors achieved an impressive power conversion efficiency (PCE) of 25.7% for small-area devices, while a 663 cm² perovskite submodule reached a PCE of 22.10% in-house and a certified PCE of 21.79%. We believe this approach holds potential for the industrialization of perovskite solar cells and aligns well with the scope of Nature Communications. Therefore, we recommend the publication of this work after minor revisions.

1. Can the authors quantify the PbI₂ deposition thickness on top of FAPbI₃ film?
2. At what temperature and vacuum conditions used to sublime Hexylammonium bromide (HABr)?
3. On page 4, line 107, the authors state: "SEM images of this 2D capped sample in Fig. 2d show that some layered structures formed on top of FAPbI₃ perovskite films with the previous pinholes disappearing, suggesting the pinhole healing effect of this deposition process." However, how did the authors confirm that the pinholes beneath the 2D capping layer were also eliminated? We suggest providing cross-sectional SEM images of the 2D perovskite-capped samples to substantiate this claim.
4. In Figure 3, the GIWAXS patterns indicate that low-dimensional perovskites grow parallel to the substrate in X = 2–4 samples. However, we are particularly interested in the spatial distribution of low-dimensional perovskites in X = 3 samples (n = 1 and n = 2). Are these phases ordered or disordered? Clarification on this point would be valuable.
5. On page 7, line 184, we noticed that the PL mapping image for X = 4 exhibits irregular regions with lower PL intensity. However, the authors do not provide an explanation for this observation in the manuscript. We recommend including a discussion to clarify the possible causes of this phenomenon.
6. In Figure 6, the authors present MPP tracking data (ISOS-L-2), which is insightful. However, we also suggest including ambient storage stability data for the unencapsulated device. This data would help demonstrate the protective effect of the 2D perovskite capping layer during storage.

Reviewer #3

(Remarks to the Author)

please refer to the attachement.

Reviewer #4

(Remarks to the Author)

Reviewer #5

(Remarks to the Author)

Version 1:

Reviewer comments:

Reviewer #1

(Remarks to the Author)

The authors have successfully addressed the reviewers' concerns. However, a few points still require attention:

1. It would be clearer to mention HABr explicitly in the introduction. The authors can refer to it directly, rather than using the more general term "organic salt", at line 68.

2. The perovskite formulation presented at line 107 applies only for $x = 2$. The authors should provide a more general formula, such as:

where x represents the molar ratio of HABr:PbI₂ = $x:1$.

Please revise accordingly.

Reviewer #2

(Remarks to the Author)

The authors have addressed our queries satisfactorily, and we recommend publishing this article in Nature Communications.

Reviewer #3

(Remarks to the Author)

Please refer to the attachment.

Reviewer #4

(Remarks to the Author)

Reviewer #5

(Remarks to the Author)

Version 2:

Reviewer comments:

Reviewer #3

(Remarks to the Author)

All issues have been resolved. It is recommended to be published in Nature Communications.

Reviewer #1 (Remarks to the Author):

I co-reviewed this manuscript with one of the reviewers who provided the listed reports as part of the Nature Communications initiative to facilitate training in peer review and appropriate recognition for co-reviewers.

In this study, the authors utilized a sequential vacuum deposition method to fabricate a 2D perovskite capping layer through a two-step evaporation process involving HABr and PbI₂. The adjustment of the molar ratios (HABr/PbI₂=x/1) influenced the dimensionality of the 2D layer, and the subsequent investigation was pivotal in determining the optimal ratio. This optimization led to a lower interfacial defect density and a reduction in recombination, leading to the best device performance observed at x = 3. Furthermore, the authors successfully upscaled the stack to a PSM with an active area of 30 × 30 cm². The work is interesting, and the achieved efficiency on the PSM is remarkable. However, similar ideas, yet with hybrid approaches, were already presented in literature (Wen, J., Zhao, Y., Wu, P. et al. Heterojunction formed via 3D-to-2D perovskite conversion for photostable wide-bandgap perovskite solar cells. *Nat Commun* 14, 7118 (2023). <https://doi.org/10.1038/s41467-023-43016-5>). In order to further consider the manuscript, the analysis performed need improvements, and the work requires major revisions.

Below are some specific comments on the manuscript:

General Comments:

a) Authors need to discuss better the novelty of the paper compared to the Wen J. et al discussed above.

Response: We thank the reviewer for the good suggestion. We have carefully checked the hybrid approach described in the literature of *Nat. Commun.*, 2023, 14, 7118 and believe that it is quite different from the hybrid one proposed in our study. In our study, the 3D perovskite was solution processed and both PbI₂ and HABr for 2D perovskite are all-vacuum-deposited, while Wen, J. et al. converted the surface of 3D perovskite/PbI₂ to quasi-2D perovskite by sequential spin-coating MAI and PEAI solutions. Such 3D-to-2D conversion has been widely applied for post-treating the surface of 3D perovskite and many works have been done on the quasi-2D or 3D to 2D conversion process (*ACS Nano* 2016, 10, 6, 5999; *Nature*, 2019, 567, 511). However, when the organic salt solution is spin-coated onto the perovskite surface as in this conversion method, it is difficult to form a uniform and thickness-controllable 2D perovskite layer on the surface (*Nat. Photonics*, 2019, 13, 460; *Chem. Rev.*, 2021, 121,

2230). As a result, it is challenging to apply this method in the fabrication process of large-area perovskite modules.

The all-vacuum deposition method in our work has the following advantages:

1. Both PbI_2 and HABr are vacuum deposited and the formed 2D perovskite layer is uniformly capped on 3D perovskite.
2. The composition, thickness, and spatial distribution of 2D perovskite can be precisely controlled by adjusting the molar ratio of PbI_2 and HABr as evidenced in Fig. 3 a-c.
3. The all-vacuum deposition method can be used to fabricate $30\text{ cm} \times 30\text{ cm}$ perovskite submodules and is capable of healing pinholes generated during the fabrication process (Fig. 4 d, e).

We posit that our research is quite different from the previous works as a novel method for fabricating passivation layer that is specifically tailored to the production of perovskite modules.

REVISED text in Introduction:

“Regularly, the 2D/3D heterojunctions are fabricated by coating the solution of organic salts, such as alkylammonium or phenylammonium halides, onto the as-fabricated 3D perovskite films or 3D perovskite/ PbI_2 films.”

“In details, during the solution based post-treatment process, the stoichiometry between PbI_2 and organic salt in the solution could dictate the formation and conversion of 2D perovskite, which could affect the interface and eventually the device performance.”

“On a solution-fabricated formamidinium lead iodide (FAPbI_3) perovskite film, all-vacuum evaporation of PbI_2 and organic salt can accurately deposit composition-tunable 2D perovskite capping layer to passivate defects and heal pinholes.”

RELATED Fig. 3a-c: GIWAXS patterns of post-treated perovskite films with HABr : $\text{PbI}_2 = x$: 1, where (a) $x=2$, (b) $x=3$ and (c) $x=4$.

RELATED Fig. 4d, e: (d) Optical microscope images of FAPbI₃ and x=3 samples. (e) Comparison on PL mapping of the FAPbI₃ and x=3 films. The curves represent the normalized PL intensity for the FAPbI₃ and x=3 samples.

b) Stability test is only limited to light soaking, however a full test at 85 °C is required considering that the use of SPIRO-OMeTAD is very limiting for thermal stress.

Response: We thank the reviewer for the kind suggestion. In this study, both the small-area solar cells and the perovskite submodules adopt the classic n-i-p structure. We agree that the inherent chemical properties of spiro-OMeTAD restrict its operation under a high temperature of 85°C (*ACS Energy Lett.*, 2017, 2, 1760; *Science*, 2022, 377, 495), whereas this HTL material is still considered the most efficient choice for the n-i-p configuration. Thus, the maximum power point (MPP) testing in our work was conducted under the temperature of 65°C, which is in accordance with the ISOS-L-2 standard. Our findings indicate that after 1000 hours of aging at 65°C, the device retains ~ 90% of its initial efficiency, demonstrating that vacuum-deposited 2D capping layer provides effective protection for the devices (Fig. 6a).

RELATED Fig. 6b: Long-term operational stability of control and vacuum devices.

c) Ensure the manuscript contains adequate references and proper contextualization within the existing literature.

Response: We thank the reviewer for this good suggestion. Our work employs an all-vacuum deposition method to fabricate 2D perovskite capping layers, which enables a good control on the preparation of passivation layers for perovskite modules. We have carefully reviewed the existing literatures and have incorporated some additional references. These references (Refs. 9, 17, 20, 34, 35, 36 and 44 in the revised manuscript) include recent studies and some foundational works that provide a comprehensive background for our findings.

REVISED and ADDED references:

9 Hu, Y. *et al.* Hybrid Perovskite/Perovskite Heterojunction Solar Cells. *ACS Nano* **10**, 5999-6007 (2016).

17 Wen, J. *et al.* Heterojunction formed via 3D-to-2D perovskite conversion for photostable wide-bandgap perovskite solar cells. *Nat. Commun.* **14**, 7118 (2023).

20 Sidhik, S. *et al.* Deterministic fabrication of 3D/2D perovskite bilayer stacks for durable and efficient solar cells. *Science* **377**, 1425-1430 (2022).

34 Jung, M.-H. Hydrophobic perovskites based on an alkylamine compound for high efficiency solar cells with improved environmental stability. *J. Mater. Chem. A* **7**, 14689-14704 (2019).

35 Lv, Y. *et al.* Hexylammonium Iodide Derived Two-Dimensional Perovskite as Interfacial Passivation Layer in Efficient Two-Dimensional/Three-Dimensional Perovskite Solar Cells. *ACS Appl. Mater. Interfaces* **12**, 698-705 (2020).

36 Lee, J.-W. *et al.* 2D perovskite stabilized phase-pure formamidinium perovskite solar cells. *Nat. Commun.* **9**, 3021 (2018).

44 Jiang, Q. *et al.* Surface passivation of perovskite film for efficient solar cells. *Nat. Photonics* **13**, 460-466 (2019).

Specific comments:

1) Line 38: The new record in the NREL chart is 27.0% (on small area, about 0.05 cm²). Please update this value.

Response: We really appreciate the reviewer for the helpful suggestion. We have updated this new record value in the revised manuscript.

REVISED text in Introduction:

The research on perovskite solar cells (PSCs) has made substantial progress as the latest certified power conversion efficiency (PCE) of small-area, single-junction PSCs has reached up to 27.0%.¹

REVISED reference:

1 Best research-cell efficiency chart. www.nrel.gov/pv/cell-efficiency.html (2025).

2) Line 86: Do the authors mean deposit with scrape-coat? Please modify.

Response: We thank the reviewer for this good suggestion. We have modified the 'scrape-coat' to 'slot-die-coat' in the revised manuscript.

REVISED text:

It usually takes less than a minute to slot-die-coat a perovskite film of 500-600 nm thickness.

3) Line 106: Please specify the chemical formula of the 2D perovskite layer for clarity.

Response: We thank the reviewer for this good suggestion. We have added the chemical formula of the 2D perovskite layer ($\text{HA}_2\text{FA}_{n-1}\text{Pb}_n\text{I}_{3n-1}\text{Br}_2$, $n=1, 2$), in the revised manuscript.

ADDED text:

The 2D perovskite capping layer was then fabricated by subsequently vacuum evaporating *n*-hexylammonium bromide (HABr) and thermal annealing (Fig. 2a), where the reaction between PbI_2 and HABr led to the formation of 2D perovskites ($\text{HA}_2\text{FA}_{n-1}\text{Pb}_n\text{I}_{3n-1}\text{Br}_2$, $n=1, 2$) with good intrinsic stability and nice surface coverage.

4) Line 116: The authors mention the evaporation of various molar ratios of HABr to PbI_2 . Did they verify these molar ratios using EDX analysis, or were they assumed based on evaporation rates/thicknesses? Please clarify.

Response: We thank the reviewer for the careful review of our manuscript. We have verified the molar ratios by controlling the evaporation thickness and the amount of materials is quantified by the signature peaks in UV-vis absorption spectra. Initially, the 7 nm PbI_2 film prepared via vacuum evaporation was dissolved in 1 mL of DMF, and the resulting solution was used to measure the absorption spectrum. By comparing this spectrum with that of PbI_2 solutions of known concentrations, the PbI_2 content in the 7

nm film was determined to be 2.5×10^{-5} mmol. Employing the same approach, the HABr content in the 6 nm film was also quantified as 5×10^{-5} mmol with a molar ratio to PbI_2 of 2:1 ($x=2$). These findings demonstrate that the material content can be precisely controlled by modulating the film thickness.

Fig. R1 (for reviewers only): (a) UV-vis spectra of 7 nm-thick PbI_2 deposited via vacuum evaporation and a solution of PbI_2 prepared at the corresponding concentration. (b) UV-vis spectra of 6 nm-thick HABr deposited via vacuum evaporation and a solution of HABr prepared at the corresponding concentration.

5) Figure 3d: The formation of the 2D perovskite layer appears non-uniform over the 3D structure. Literature has shown that 2D perovskites can either form a conformal coating over the underlying layer or exhibit discontinuities. Please address this point explicitly and clarify the reason behind this significant lack of uniformity.

Response: We thank the reviewer for the helpful question. The observed non-uniformity of 2D perovskite layers can be attributed to our TEM sample preparation methodology. Specifically, the 2D/3D film was scratched off from the substrate and dispersed in diethyl ether, which was dropped on a copper grid (*Nat. Commun.*, 2018, 9, 3021). This preparation process may potentially lead to discontinuous sample characteristics. From the cross-sectional SEM image of the 2D capped sample (Supplementary Fig. 2), it can be observed that the 2D perovskite is uniformly distributed on the 3D perovskite.

ADDED Supplementary Fig. 2: Cross-sectional SEM image of 2D capped sample.

6) Ensure that the sources or databases used for obtaining the XRD peak references are clearly specified.

Response: We thank the reviewer for this good suggestion. We have confirmed the accuracy of the XRD peak positions by comparing our data with the previously reported XRD patterns of HA-based 2D perovskites (*Energy Environ. Sci.*, 2019, 12, 2192; *J. Mater. Chem. A*, 2019, 7, 14689; *ACS Appl. Mater. Interfaces*, 2020, 12, 698), which have been cited in the manuscript as Refs. 14, 34, 35.

7) In Supplementary Fig. 7, the XRD spectra for samples with 2D capping layers across various annealing durations are presented. The final two annealing times for $x=2$ and $x=4$ exhibit similar spectra, indicating a stable phase of the 2D has been achieved. However, for $x=3$, some changes occur in the plot between 60 and 90 seconds. Furthermore, the authors state in the experimental section that devices were annealed for 2 minutes. Therefore, the XRD pattern does not directly correlate with the conditions used for device fabrication. It is advised to include the XRD pattern of up to 2 minutes for $x=3$.

Response: We thank the reviewer for this good suggestion. We conducted re-examinations of the $x=3$ and $x=4$ samples annealed for 120 seconds using XRD analysis and subsequently reconstructed the corresponding XRD patterns. The results are presented in the updated Supplementary Fig. 8.

REVISED Supplementary Fig. 8: XRD patterns of (a) $x=2$, (b) $x=3$ and (c) $x=4$ samples with different annealing times.

8) Line 215-216: The poor performance of the device with $x=4$ is attributed to excessive thickness of the 2D capping layer in this case. However, the previous analysis has shown a lower electric field, with a less intense peak compared with $x=3$, at the PVK/HTL interface. This analysis also showed reduced uniformity and higher surface potential. The authors should consider and correlate with the previous analysis when discussing the electrical characteristics.

Response: We thank the reviewer for this good suggestion. It has also been demonstrated that $n=2$ 2D perovskites can form a Type-I band alignment with FAPbI_3 perovskite (*Science*, 2022, 377, 1425), which is consistent with the results obtained from our in-depth analysis of ultraviolet photoelectron spectroscopy (UPS). Furthermore, it has been proposed that the thickness of 2D perovskites forming a Type-I band alignment should generally be less than 20 nm (*Science*, 2022, 377, 1425). An excessively thick 2D perovskite layer can hinder hole transport, leading to a significant reduction in device efficiency. The poor performance of $x=4$ sample is consistent with that of the 50 mM HABr treated device in the literature (*Nat. Energy*, 2024, 9, 457).

ADDED text:

However, for the $x=4$ sample, V_{oc} , J_{sc} and FF of the device have drastically decreased. The vacuum deposited $n=2$ 2D perovskite layer, while demonstrating effective passivation of surface defects in perovskite materials, exhibits a detrimental impact on device performance due to excessive overlayer thickness, which significantly impedes hole transport and consequently leads to a substantial reduction in device efficiency.²⁰

ADDED reference:

20. Sidhik, S. *et al.* Deterministic fabrication of 3D/2D perovskite bilayer stacks for durable and efficient solar cells. *Science* **377**, 1425-1430 (2022).

9) Line 238: A lower ideality factor is not always indicative of improved device

performance, as it depends on the nature of the recombination processes occurring in the device (doi: 10.1002/aenm.202000502). Please consider that.

Response: We thank the reviewer for this good suggestion. We have modified the relevant discussions to enhance the academic rigor and precision.

REVISED text:

Fig. 5c illustrates that vacuum device displays a lower ideal factor (n) value and series resistance (R_s). Considering that the EQE_{EL} have confirmed that the interface recombination is largely suppressed, the lower n value demonstrates a lower recombination process in the vacuum devices.^{44,46}

ADDED reference:

44 Jiang, Q. *et al.* Surface passivation of perovskite film for efficient solar cells. *Nat. Photonics* **13**, 460-466 (2019).

46 Caprioglio, P. *et al.* On the Origin of the Ideality Factor in Perovskite Solar Cells. *Adv. Energy Mater.* **10**, 2000502 (2020).

10) Line 263: Correct the figure reference from "Fig. 4f" to "Fig. 5f".

Response: We thank the reviewer for the careful review of our manuscript. We have already corrected Fig. 4f to Fig. 5f.

11) Lines 264-265: The discussion on energy level alignment is fragmented, continuing onto line 269. It would be more coherent to consolidate this section into a unified discussion. Additionally, it is recommended to include UPS measurements for the control device for a more comprehensive analysis.

Response: We thank the reviewer for this good suggestion. We have utilized the depth-independent UPS measurement for the control sample and have made a more comprehensive analysis. We have revised the corresponding descriptions in the manuscript to ensure that the content regarding the UPS test results is presented in a more structured and comprehensive manner.

REVISED text:

The compositional gradients and surface energetics can be precisely tuned. In contrast, the VBM in the control sample did not show significant changes, as shown in Supplementary Fig. 25 and Table 4. Moreover, the ideal energy level alignment allows

for efficient charge transfer at the hole-selective heterointerface, while the shallower conduction band minimum (CBM) of vacuum sample can block electron transport and thus reduce electron-hole recombination at the interface.

ADDED Supplementary Fig. 25: Depth-dependent ultraviolet photoemission spectroscopy (UPS) (a) at the secondary cut off energy for control sample. (b) at the valence band maximum for control sample.

ADDED Supplementary Table 4: Corresponding parameters of control sample calculated from depth-dependent UPS measurements.

Etching time (s)	VB (eV)	E_{cutoff} (eV)	WF (eV)	VBM (eV)
0	0.92	16.14	-5.08	-6.00
300	0.88	16.13	-5.09	-5.97
600	0.88	16.14	-5.08	-5.96

12) Line 266: The conclusions drawn from the XPS analysis are unclear. Please clarify the key findings and their significance.

Response: We thank the reviewer for this good suggestion. In the corresponding section of the manuscript, we have provided a more comprehensive and detailed discussion on the conclusions derived from the XPS analysis. As illustrated in Supplementary Fig. 26, the atomic percentage of C in the thin film from surface to the bulk exhibits a notable decrease from 50% to 20%, indicating a gradient distribution pattern of 2D perovskite. Within the bulk phase of the perovskite, the atomic ratio of C to I approximates 1:3, suggesting that the bulk primarily consists of FAPbI₃ perovskite.

ADDED text:

As illustrated in Supplementary Fig. 26, depth-dependent X-ray photoemission

spectroscopy (XPS) shows that the atomic percentage of C in the thin film from surface to the bulk exhibits a notable decrease from 50% to 20%, indicating a gradient distribution pattern of 2D perovskite. Within the bulk phase of the perovskite, the atomic ratio of C to I is approximately 1:3, suggesting that the bulk primarily consists of FAPbI₃ perovskite.

13) The inset in Supplementary Fig.27 is not clearly visible and indistinguishable. Please improve its visibility and clarity.

Response: We thank the reviewer for the careful review of our manuscript. In the captions of the figure, we have incorporated comprehensive descriptions of the illustrations to enhance the clarity and precision of the content.

REVISED Supplementary Fig. 27: Moisture stability of control and vacuum samples when stored under 80% RH at 25°C in ambient air for 1 day. The inset image is FAPbI₃ perovskite thin films with and without a 2D capping layer after being stored under 80% RH at 25°C in ambient air for 1 day.

REVISED text:

After being stored under 80% relative humidity (RH), 25 °C in ambient conditions for one day, the control sample had undergone the phase change from α - to δ -FAPbI₃ as shown in Supplementary Fig. 27. In contrast, the vacuum area covered with 2D perovskite capping layer remained black phase, which revealed that the 2D layer exhibited excellent protection to the FAPbI₃ perovskite.

14) Line 299: Specify large area since slot-die coating was employed only for module

fabrication.

Response: We thank the reviewer for the good suggestion. We have modified the description in the revised manuscript.

REVISED text:

This strategy combines the rapid preparation of 3D perovskite film by solution method (slot-die coating for module fabrication) and precise deposition of 2D perovskite passivation layer by vacuum evaporation.

15) Line 31: $30 \times 30 \text{ cm}^2$ is a small module size, not a submodule, as reported in doi: 10.1002/pip.3489. Please modify that.

Response: We thank the reviewer for the careful review of our manuscript. In this article (*Prog. Photovolt. Res. Appl.*, 2022, 30, 360.), a module with an aperture area exceeding 800 cm^2 is defined as a small module. The aperture area of our module fabricated on $30 \times 30 \text{ cm}^2$ substrate is 663 cm^2 , thus it is classified as a submodule.

Reviewer #2 (Remarks to the Author):

In this manuscript, Fan et al. reports an efficient and precise solution-vacuum deposition strategy that enables the rapid and controlled fabrication of 2D/3D perovskite submodules. The vacuum-deposited 2D capping layers can be finely tuned by adjusting the molar ratio of PbI_2 and HABr . As a result, the authors achieved an impressive power conversion efficiency (PCE) of 25.7% for small-area devices, while a 663 cm^2 ; perovskite submodule reached a PCE of 22.10% in-house and a certified PCE of 21.79%. We believe this approach holds potential for the industrialization of perovskite solar cells and aligns well with the scope of Nature Communications. Therefore, we recommend the publication of this work after minor revisions.

1 Can the authors quantify the PbI_2 deposition thickness on top of FAPbI_3 film?

Response: We thank the reviewer for the careful read of our manuscript and the good suggestion. As illustrated in Fig. R2 (for reviewers only), the TEM image clearly reveals the presence of the PbI_2 layer, which was measured to be approximately 7 nm in thickness, consistent with the value indicated by the evaporation deposition system.

Fig. R2 (for reviewers only): TEM image of FAPbI_3 - PbI_2 sample.

2. At what temperature and vacuum conditions used to sublime Hexylammonium bromide (HABr)?

Response: We thank the reviewer for this good question. The Hexylammonium bromide (HABr) was evaporated under pressure of less than 10^{-3} Pa. The voltage of the evaporation boat is 8 V and HABr was evaporated by improving the current to 12 A.

3. On page 4, line 107, the authors state: "SEM images of this 2D capped sample in Fig. 2d show that some layered structures formed on top of FAPbI_3 ; perovskite films with

the previous pinholes disappearing, suggesting the pinhole healing effect of this deposition process." However, how did the authors confirm that the pinholes beneath the 2D capping layer were also eliminated? We suggest providing cross-sectional SEM images of the 2D perovskite-capped samples to substantiate this claim.

Response: We thank the reviewer for this helpful suggestion. In the supporting information, we have supplemented the cross-sectional SEM image of the 2D capped sample, and corresponding descriptions have been added to the manuscript.

ADDED text:

SEM images of this 2D capped sample in Fig. 2d show that some layered structures formed on top of FAPbI₃ perovskite films with the previous pinholes disappeared (Supplementary Fig. 2), suggesting the pinhole-healing effect of this deposition process.

ADDED Supplementary Fig. 2: Cross-section SEM image of 2D capped sample.

4. In Figure 3, the GIWAXS patterns indicate that low-dimensional perovskites grow parallel to the substrate in $x = 2,3$ and 4 samples. However, we are particularly interested in the spatial distribution of low-dimensional perovskites in $x = 3$ samples ($n = 1$ and $n = 2$). Are these phases ordered or disordered? Clarification on this point would be valuable.

Response: We thank the reviewer for the careful review of our manuscript. We posit that in the $x=3$ sample, the $n=1$ and $n=2$ 2D perovskite layers are sequentially arranged from top to bottom in conjunction with 3D perovskites. As evidenced by the in situ GIWAXS patterns presented in Fig. 3 (e), the initial formation of $n=1$ 2D perovskite occurs at the perovskite surface. Prolonged annealing facilitates the reaction between the $n=1$ 2D perovskite at the interface and the 3D perovskite, resulting in the generation of $n=2$ lower-dimensional perovskite. Furthermore, the cross-sectional KPFM images

depicted in Fig. R3 (for reviewers only) corroborate the continuous distribution of 2D perovskite across the surface of the 3D perovskite.

RELATED Fig. 3e: *in-situ* GIWAXS patterns of x=3 sample.

Fig. R3 (for reviewers only): Cross-sectional KPFM image of x=3 sample.

5. On page 7, line 184, we noticed that the PL mapping image for $x = 4$ exhibits irregular regions with lower PL intensity. However, the authors do not provide an explanation for this observation in the manuscript. We recommend including a discussion to clarify the possible causes of this phenomenon.

Response: We thank the reviewer for this good suggestion. We consider that the irregular regions exhibiting lower photoluminescence (PL) intensity in the PL mapping image at $x=4$ correspond to uneven morphology of the generated $n=2$ 2D perovskite. As shown in Supplementary Fig. 7c, due to the excessive introduction of HABr in $x=4$ sample, the generated 2D perovskite has made the film rougher, resulting in irregular protrusions or depressions, which leads to an inhomogeneous distribution of PL intensity. The corresponding discussions are modified in the revised manuscript.

RELATED Supplementary Fig. 7c: Top-view SEM images of $x=4$ sample.

ADDED text:

In the x=4 sample, irregular areas with lower PL intensity were observed, which corresponds to the rougher 2D perovskite capping layer (Supplementary Fig. 7c).

6. In Figure 6, the authors present MPP tracking data (ISOS-L-2), which is insightful. However, we also suggest including ambient storage stability data for the unencapsulated device. This data would help demonstrate the protective effect of the 2D perovskite capping layer during storage.

Response: We thank the reviewer for this good suggestion. We have incorporated storage stability data along with pertinent descriptive analyses in the revised manuscript.

REVISED Supplementary Fig. 28: The shelf-life stability of unencapsulated control and vacuum PSCs.

ADDED text:

For device stability, the shelf life was measured by storing the unencapsulated devices in dark at 25 °C and in an ambient air glovebox with 25% RH. Supplementary Fig. 28 demonstrates a 20 % decrease in the PCE of the control device after 1000 h of aging (ISOS-D-1), while the vacuum device retains 95.5 % of its initial PCE.

Reviewer #3 (Remarks to the Author):

In this manuscript, Fan et al. have developed a solution-vacuum hybrid batch fabrication strategy, which could precisely deposit nanoscale two-dimensional (2D) capping layer via vacuum evaporation on a solution deposited three-dimensional (3D) bulk film. Based on this method, the as fabricated 30 cm × 30 cm pinhole-free perovskite submodule achieved a champion PCE up to 22.10% (certified PCE of 21.79%) of 663 cm² aperture area demonstrating exceptional scalability in terms of process amplification. Overall, this work provides a potential method for preparing high efficiency modules in industrial production. I would like to recommend this work be published in Nature Communications after some minor revisions.

1. In page 3, line 78, in Fig. 1, In my assessment, the methodology pertaining to the electrode fabrication in the orange region warrants further scrutiny, as the figure omits the step of electrode deposition via thermal evaporation.

Response: We thank the reviewer for the careful read of our manuscript and the good suggestion. We have made modifications to the electrode deposition section in Fig. 1's flowchart to enhance its alignment with the subheading.

REVISED Fig 1: Schematics on the proposed solution-vacuum hybrid perovskite submodule manufacturing process. FTO: fluorine-doped tin oxide, ETL: electron transport layer and HTL: hole transport layer.

2. In Supplementary Fig. 10, I am interested in the types of low dimensional perovskite in the x=3 sample. In JACS, 2023, 145, 8209, the authors mentioned that HABr could

react with FAPbI₃ and PbI₂. So how did the author determine the composition of the low-dimensional perovskite in the x=3 sample? In other words, would HABr react with FAPbI₃ in the x=3 sample?

Response: We thank the reviewer for the careful review of our manuscript. In our previous work, it has been mentioned that HABr reacts with PbI₂ and FAPbI₃, respectively. In this work, we observed that at the interface between 3D perovskite and 2D perovskite, HA₂PbI₂Br₂ reacts with FAPbI₃ to form n=2 2D perovskite. As illustrated in Supplementary Fig. 6, a distinct characteristic peak at 14.2° is evident in the sample with x=4. By comparison with Supplementary Fig. 5, this characteristic peak corresponds to the (111) lattice plane of the n=2 perovskite, which is consistent with reports (*Energy Environ. Sci.*, 2019, 12, 2192.). Therefore, we hypothesize that the reaction of HABr with PbI₂ and FAPbI₃ predominantly generates n=1 and n=2 2D perovskites in the x=3 sample.

RELATED Supplementary Fig. 5: XRD patterns of HA₂PbI₂Br₂ (n=1) and HA₂FAPb₂I₅Br₂ (n=2) perovskite films.

RELATED Supplementary Fig. 6: XRD patterns of perovskite films with HABr: PbI₂ =x:1, x is 0, 2, 3 and 4, respectively.

3. In page 8, line 223, In the device performance part, the authors do not give data for steady state output. In my opinion, the steady-state output is an important index to measure the efficiency and stability of the device in the process of testing unencapsulated devices

Response: We thank the reviewer for this good suggestion. Supplementary steady-state output measurements were performed, with the results detailed in Supplementary Fig. 18.

ADDED text: Steady-state output results demonstrate that the 2D capping layer exhibits excellent protective performance under continuous illumination operating conditions of the device, as shown in Supplementary Fig. 18.

ADDED Supplementary Fig. 18: Steady-state output test of the control and vacuum devices.

4. In Figure 3, can the author provide information on what 2D structures the signals

appearing in the GIWAXS results belong to when q_z is between 5 and 10? Is it the second order of $n=2$ and $n=1$?

Response: We thank the reviewer for this good suggestion. The signals appearing in the GIWAXS results belong to when q_z is between 5 and 10 is the second order of $n=2$ and $n=1$. We have indexed the lattice in Fig. R4.

Fig. R4 (for reviewers only): GIWAXS patterns of post-treated perovskite films with HABr: $\text{PbI}_2 = x: 1$, where (a) $x=2$, (b) $x=3$ and (c) $x=4$.

5. I am curious about the maximum area that the author has tried to prepare using this method? Can the pinhole-free effect be maintained under the maximum production area?

Response: We thank the reviewer for the careful review of our manuscript. Under the current technical conditions of our laboratory, the maximum area of perovskite thin films that can be produced is $30 \text{ cm} \times 30 \text{ cm}$. Our experimental results have not revealed any significant presence of pinholes to date. We believe that this methodology holds the potential for the fabrication of large perovskite solar modules

6. I think if conditions permit, the author can measure TAS to more intuitively illustrate the energy transfer rate of carriers in a quasi-two-dimensional system under optimal conditions. It can also prove that reducing defects reduces non-radiative recombination, making the mechanism analysis of the article more in-depth and direct.

Response: We thank the reviewer for this good suggestion. We sincerely apologize that, due to the current limitations of our laboratory facilities, we are unable to conduct transient absorption spectroscopy (TAS) measurements at this stage. Through KPFM characterization, we have demonstrated the optimal energy transfer rate of charge carriers and the reduction of non-radiative recombination in quasi-2D systems. The planar KPFM images reveal that the sample with $x=3$ exhibits the lowest surface potential, which facilitates the hole transfer from the 3D perovskite to the hole transport

layer (HTL). Furthermore, the surface potential profile of the x=3 sample is more uniform and lower compared to those of the x=2 and x=4 samples, indicating effective suppression of recombination at the perovskite film surface and grain boundaries. From the cross-section KPFM images and PL mapping measurements, it can be inferred that defects in the x=3 sample are passivated, leading to a reduction in non-radiative recombination.

7. I suggest that the author should review Materials and Methods carefully to see if there are any errors. For example, in line 412, is DMF/DMSO 8:1 or 8:2?

Response: We thank the reviewer for the helpful question. We have meticulously reviewed and revised the descriptions pertaining to the Materials and Methods part. It is important to clarify that the solvent ratio employed in our study was indeed 1 mL DMF: DMSO at a ratio of 8:1 (889 μ L DMF and 111 μ L DMSO), as referenced in the previous work (*Nature*, 2023, 616, 724.). We have also modified the description in the revised manuscript.

ADDED text: The 1.5 M FAPbI₃ perovskite precursor was prepared by mixing 1.5 mmol PbI₂, 1.5 mmol FAI and 0.525 mmol MACl in 1 mL mix solvent (889 μ L DMF and 111 μ L DMSO).

Reviewer #4 (Remarks to the Author):

Response: We thank the reviewer for the time spent on evaluating our manuscript and the valuable suggestions.

Reviewer #5 (Remarks to the Author):

Response: We thank the reviewer for the time spent on evaluating our manuscript and the valuable suggestions.

Reviewer #1 (Remarks to the Author):

The authors have successfully addressed the reviewers' concerns. However, a few points still require attention:

1. It would be clearer to mention HABr explicitly in the introduction. The authors can refer to it directly, rather than using the more general term "organic salt", at line 68.

Response: We thank the reviewer for the good suggestion. We have changed the 'organic salt' to '*n*-hexylammonium bromide (HABr)' in the revised manuscript.

2. The perovskite formulation presented at line 107 applies only for $x = 2$. The authors should provide a more general formula, such as: $\text{HA}_x\text{FA}_{n-1}\text{Pb}_n\text{I}_{3n-1}\text{Br}_x$, where x represents the molar ratio of $\text{HABr}:\text{PbI}_2 = x:1$. Please revise accordingly.

Response: We thank the reviewer for the good suggestion. We have modified the description of the 2D perovskite formula in the revised manuscript.

REVISED text:

The reaction between HABr and PbI_2 with different molar ratios ($\text{HABr}:\text{PbI}_2=x:1$, $x=2, 3$ and 4) will lead to the formation of different 2D perovskites, such as $\text{HA}_2\text{PbI}_2\text{Br}_2$ and $\text{HA}_2\text{FAPb}_2\text{I}_5\text{Br}_2$ with good intrinsic stability and nice surface coverage.

Reviewer #2 (Remarks to the Author):

The authors have addressed our queries satisfactorily, and we recommend publishing this article in Nature Communications.

Response: We thank the reviewer for the positive comment.

Reviewer #3 (Remarks to the Author):

The author has basically addressed the questions I asked before, but through the questions from other reviewers and the author's answers, I still have a few questions I would like to clarify.

1. In the *in situ* GIWAXS results presented in Figure 3e of the article, it is observed that the $n=1$ phase forms prior to the $n=2$ phase. This observation is intriguing, as the theoretical analysis suggests that the formation energy of $n=2$ should be lower than that of $n=1$. The authors are encouraged to elucidate the underlying reasons for this phenomenon. While the process is discussed in Figure 11 of the supplementary information, an explicit explanation for the precedence of $n=1$ over $n=2$ is lacking. It may be beneficial for the authors to incorporate simple density functional theory (DFT) formation energy calculations to further analyze and clarify this occurrence. Or can the author provide some explanation through other relevant literature?

Response: We thank the reviewer for the helpful suggestion. When HABr, FAI, and PbI_2 are dissolved in a solution and spin-coated to form a thin film, the low-dimensional perovskite with $n=2$ will be preferentially formed. While here in our work, the fabrication method and thus the reaction route are different. In detail, we deposited a layer of PbI_2 on the surface of the 3D perovskite, followed by the deposition of a layer of HABr. As a result, HABr will react with PbI_2 to form $n=1$ 2D perovskite first. Some of the residual HABr will then promote the reaction from $n=1$ to $n=2$ 2D perovskite. This is precisely what is depicted in the *in-situ* GIWAXS profile shown in Figure 3e. In a very recent work (*Science*, 2025, 388, 639. **Fig. 1A as attached below**), Tan et al. used octylammonium bromide (OABr) to post-treat the 3D perovskite film with excess PbI_2 . The formation of the $n=1$ 2D perovskite forms prior to $n=2$ was also observed, as shown in **Fig R1**.

Fig. R1 from *Science*, 2025, 388, 639. (**Fig. 1A**)

2. In this work, the authors conducted extensive structural characterization; however, photoluminescence (PL) tests for samples with $x=1$, 2, and 3 were not performed (normal PL measurement, not PL mapping). To strengthen their findings, the authors should provide further evidence through PL tests demonstrating that the samples predominantly exhibit phases with $n=1$ and/or $n=2$. Theoretically, phases with $n=1$, 2, and 3 should coexist, but the formation of certain two-dimensional phases is minimal, making them challenging to detect using X-ray diffraction (XRD) or grazing incidence wide-angle X-ray scattering (GIWAXS).

Response: We thank the reviewer for the careful review of our manuscript. In the revised manuscript, we have supplemented the PL spectra (**Supplementary Fig. 8**). It is observable that different 2D perovskite phases exist in the samples with $x=2$, 3 and 4. We have revised the corresponding descriptions and figure in the manuscript and supplementary information, respectively.

REVISED text: Hence, the exact chemical composition and n value of the 2D perovskites can be precisely controlled by adjusting the ratio of HABr to PbI_2 , which was further corroborated by the crystal structures identified in the HR-TEM images (Fig. 3d) and PL spectra (Supplementary Fig. 8) for these samples.

REVISED Supplementary Fig. 8: PL spectra of FAPbI_3 perovskite films with HABr: $\text{PbI}_2 = x:1$, x is 2, 3 and 4, respectively.

3. In the latest submission, the author presents a scanning electron microscopy (SEM) cross-section of the 2D capping. However, to facilitate a comprehensive comparison, it

is imperative that the author includes a comparative analysis of the samples before and after the 2D treatment. Furthermore, the provided cross-section indicates that the surface post-treatment exhibits considerable roughness and is not entirely smooth. The author is encouraged to elucidate the underlying reasons for this observation, potentially through comparative illustrations.

Response: We thank the reviewer for the careful review of our manuscript. We have added the cross-sectional SEM of the FAPbI₃ film without 2D capping in **Supplementary Fig. 2a**, where the surface of FAPbI₃ sample is also a little rough. This result is consistent with similar roughness as profiled by AFM measurement (**Supplementary Fig. 3**). It is suspected that such a rough surface may be caused by the rough FTO substrate. Similarly rough surface can also be observed in other reports (*Nat. Nanotechnol.* 2022, 17, 598; *Nat. Energy*, 2024, 9, 316; *Nat. Energy*, 2024, 9, 1540.). We have revised the corresponding discussions in the manuscript and supplementary information, respectively.

REVISED text: The cross-sectional SEM images and atomic force microscopy (AFM) images reveal that both the FAPbI₃ and 2D capped samples exhibit similar roughness (Supplementary Fig. 2 and Supplementary Fig. 3), indicating that the 2D capping layer has a good coverage on 3D FAPbI₃ films.

REVISED Supplementary Fig. 2: Cross-sectional SEM images of (a) FAPbI₃ and (b) 2D capped samples.

REVISED Supplementary Fig. 3: AFM images of (a) FAPbI₃ and (b) 2D capped samples.

4. In Fig. R4, we can see that the sample with $x=2$ has only the phases of (002) and (004) with $n=1$, among which (004) is very weak, which proves that a single layer of $n=1$ phase is formed. When $x=3$, we see (002), (004) and (006) with $n=2$. I am curious why the sample with $x=3$ has a better effect, because from a structural point of view, for post-processing, the stacking of too many two-dimensional phases (that is, (002), (004), (006) appear at the same time and the signal is very strong) is not friendly to charge transport. On the contrary, the two-dimensional phase formed on the surface of the sample with $x=1$ is mostly the first order, and the second order with very weak signal indicates that there is no excessive stacking of two-dimensional structures, which should be conducive to charge transport and easier to achieve the effect of electron tunneling. Therefore, can the author explain why the sample with $x=3$ is better?

Response: We thank the reviewer for the insightful question. Besides the stacking thickness of 2D perovskite layers, the energy-level alignment is another key factor that influences the charge transport process. Some previous works have reported that 2D perovskites with some higher n values tend to form Type-II band alignment, which is helpful for charge transport (*Science*, 2022, 376, 73; *Science*, 2022, 377, 1425; *Nat. Energy*, 2024, 9, 457). Our results have also demonstrated that the $n=1$ 2D perovskite forms a type-I band alignment with 3D perovskite, which implies that holes must tunnel through the $n=1$ two-dimensional perovskite layer to reach hole transport layer. For the sample with $x=3$, the 2D perovskite with both $n=1$ and $n=2$ can form a Type-II band alignment (**Supplementary Fig.26 and Fig. 5f**), which is conducive to the extraction and transport of holes. The effect of thicker 2D perovskite layer to hinder hole transport becomes obvious when $x=4$, leading to a significant reduction in device efficiency. Therefore, the sample with $x=3$ is comparably the best performing one.

Reviewer #4 (Remarks to the Author):

Response: We thank the reviewer for the time spent on evaluating our manuscript and the valuable suggestions.

Reviewer #5 (Remarks to the Author):

Response: We thank the reviewer for the time spent on evaluating our manuscript and the valuable suggestions.

Reviewer #3 (Remarks to the Author):

All issues have been resolved. It is recommended to be published in Nature Communications.

Response: We thank the reviewer for the positive comment.

In this manuscript, Fan et al. have developed a solution-vacuum hybrid batch fabrication strategy, which could precisely deposit nanoscale two-dimensional (2D) capping layer via vacuum evaporation on a solution deposited three-dimensional (3D) bulk film. Based on this method, the as-fabricated 30 cm × 30 cm pinhole-free perovskite submodule achieved a champion PCE up to 22.10% (certified PCE of 21.79%) of 663 cm² aperture area demonstrating exceptional scalability in terms of process amplification. Overall, this work provides a potential method for preparing high efficiency modules in industrial production. I would like to recommend this work be published in Nature Communications after some minor revisions.

1. In page 3, line 78, in Fig. 1, In my assessment, the methodology pertaining to the electrode fabrication in the orange region warrants further scrutiny, as the figure omits the step of electrode deposition via thermal evaporation.
2. In Supplementary Fig. 10, I am interested in the types of the low dimensional perovskite in the X=3 sample. In JACS, 2023, 145, 8209, the authors mentioned that HABr could react with FAPbI₃ and PbI₂. So how did the author determine the composition of the low-dimensional perovskite in the x=3 sample? In other words, would HABr react with FAPbI₃ in the x=3 sample?
3. In page 8, line 223, In the device performance part, the authors do not give data for steady-state output. In my opinion, the steady-state output is an important index to measure the efficiency and stability of the device in the process of testing unencapsulated devices
4. In Figure 3, can the author provide information on what 2D structures the signals appearing in the GIWAXS results belong to when q_z is between 5 and 10? Is it the second order of n=2 and n=1?
5. I am curious about the maximum area that the author has tried to prepare using this method? Can the pinhole-free effect be maintained under the maximum production area?
6. I think if conditions permit, the author can measure TAS to more intuitively illustrate the energy transfer rate of carriers in a quasi-two-dimensional system under optimal conditions. It can also prove that reducing defects reduces non-radiative recombination, making the mechanism analysis of the article more in-depth and direct.
7. I suggest that the author should review Materials and Methods carefully to see if there are any errors. For example, in line 412, is DMF/DMSO 8:1 or 8:2?

The author has basically addressed the questions I asked before, but through the questions from other reviewers and the author's answers, I still have a few questions I would like to clarify.

1. In the in situ GIWAXS results presented in Figure 3e of the article, it is observed that the $n=1$ phase forms prior to the $n=2$ phase. This observation is intriguing, as the theoretical analysis suggests that the formation energy of $n=2$ should be lower than that of $n=1$. The authors are encouraged to elucidate the underlying reasons for this phenomenon. While the process is discussed in Figure 11 of the supplementary information, an explicit explanation for the precedence of $n=1$ over $n=2$ is lacking. It may be beneficial for the authors to incorporate simple density functional theory (DFT) formation energy calculations to further analyze and clarify this occurrence. Or can the author provide some explanation through other relevant literature?

2. In this work, the authors conducted extensive structural characterization; however, photoluminescence (PL) tests for samples with $x=1, 2,$ and 3 were not performed (normal PL measurement, not PL mapping). To strengthen their findings, the authors should provide further evidence through PL tests demonstrating that the samples predominantly exhibit phases with $n=1$ and/or $n=2$. Theoretically, phases with $n=1, 2,$ and 3 should coexist, but the formation of certain two-dimensional phases is minimal, making them challenging to detect using X-ray diffraction (XRD) or grazing incidence wide-angle X-ray scattering (GIWAXS).

3. In the latest submission, the author presents a scanning electron microscopy (SEM) cross-section of the 2D capping. However, to facilitate a comprehensive comparison, it is imperative that the author includes a comparative analysis of the samples before and after the 2D treatment. Furthermore, the provided cross-section indicates that the surface post-treatment exhibits considerable roughness and is not entirely smooth. The author is encouraged to elucidate the underlying reasons for this observation, potentially through comparative illustrations.

4. In Fig. R4, we can see that the sample with $x=2$ has only the phases of (002) and (004) with $n=1$, among which (004) is very weak, which proves that a single layer of

$n=1$ phase is formed. When $x=3$, we see (002), (004) and (006) with $n = 2$. I am curious why the sample with $x=3$ has a better effect, because from a structural point of view, for post-processing, the stacking of too many two-dimensional phases (that is, (002), (004), (006) appear at the same time and the signal is very strong) is not friendly to charge transport. On the contrary, the two-dimensional phase formed on the surface of the sample with $x=1$ is mostly the first order, and the second order with very weak signal indicates that there is no excessive stacking of two-dimensional structures, which should be conducive to charge transport and easier to achieve the effect of electron tunneling. Therefore, can the author explain why the sample with $x=3$ is better?